# Pre-clinical evaluation of a low-cost tool for skin temperature measurements as a proxy to assess autonomic nerve function in leprosy neuropathy

Arjan J. Knulst[1,2]*, Fleur van den Bogaert[2], Alexander Kuipers[2], Lieke Roelofs[2], Jenny Dankelman[2], Wim Brandsma[3], Corine Knulst-Verlaan[1], Suraj Maharjan[1]

**1** INF Green Pastures Hospital and Rehabilitation Center, Pokhara, Nepal, **2** Department of Biomechanical Engineering, Delft University of Technology, Delft, The Netherlands, **3** Independent Leprosy Rehabilitation Consultant, Hoevelaken, The Netherlands

* a.j.knulst@tudelft.nl

## Abstract

Peripheral autonomic nerve function (ANF) impairment (ANFI) can be one of the first indicators for leprosy or leprosy neuropathy. However, within leprosy, hardly any ANF assessment methods are used in current practice. Skin temperature could be a proxy measure to assess ANF. Therefore, this research aims to explore whether low-cost infrared (IR) video thermography can be used as an ANF assessment tool by measuring the skin temperature response (STR) of human hands before, and after applying a cold pressor test (CPT). A protocol was defined to perform a baseline measurement, apply CPT, and record the resulting STR curve during 15 minutes. An IR video camera connected to a mobile phone was selected as sensor. A setup was developed to immobilize the position of the hands and fingers relative to the camera. A Python script was developed to extract the hand palm skin temperature STR curve from an IR video for 12 ulnar and median innervated regions of interest (ROI) in 1s intervals. A Matlab script was developed to post-process the raw temperature data into filtered data. This data is used to calculate key metrics that describe the STR curve. This approach was evaluated on technical accuracy and precision by comparing IR data for 3 cameras to a reference sensor. The variability caused by the observer analyzing the data was studied by analyzing the same video 5 times by 2 observers. The subject variability was studied by enrolling 7 subjects into a pilot, testing them daily for 5 consecutive days. The results show a high mean Interclass correlation of 0.94 between the 3 IR cameras and the reference sensor. Bland-Altman plots show a mean accuracy of +0.090°C between the cameras and the reference, and a variation between −1.30 and +1.50°C. High agreement was shown between observers analyzing the data. The pilot test showed high variability in STR curve within subjects. Although the general shape of the STR was similar, the location of the steep increase in recovery varied strongly within and between subjects.

**Data availability statement:** The data is available from Mendeley Data: https://data.mendeley.com/datasets/y8rjcspc29/1.

**Funding:** This study was part of a research project funded by the Leprosy Research Initiative (https://www.leprosyresearch.org/), the Netherlands (Research Capacity Strengthening Grant FP23_CS.6), awarded to dr S. Maharjan. The funder did not have any influence design or outcome of the reported manuscript.

**Competing interests:** The authors have declared that no competing interests exist.

This study shows that a low-cost, portable IR camera can be used to measure STR of human hands after CPT. A pilot study showed high subject variability for repeated testing of the STR curve. Future research is needed to establish its value in assessing ANF in leprosy patients or other systemic and local neuropathies and traumatic nerve conditions.

## Introduction

Leprosy is an infectious disease that affects skin and superficial peripheral nerves of approximately 200.000 people annually worldwide [1–4]. Diagnosis is mainly based on three cardinal signs: skin smear, typical skin lesions and enlarged peripheral nerve(s) [3,5]. Additionally, nerve function tests are used to supplement the clinical diagnosis. Currently, nerve function tests are limited to Voluntary Muscle Testing (VMT) and Sensory Testing (ST) [6–9]. Some research has been done on tests to assess the peripheral autonomic nerve function (ANF), but these are hardly used in practice [10–12].

However, it has been suggested that peripheral ANF Impairment (ANFI) can be one of the first indicators for leprosy or leprosy neuropathy [11,13–16]. Also, ANF might be affected without showing VMT or ST impairments. Peripheral ANF controls, for instance, skin temperature and sweating. It might therefore be possible to use Skin Temperature Response (STR) as a proxy measure for assessing peripheral ANF. Various studies [17–22] have been done that found deviating skin temperatures in leprosy or diabetic patients having peripheral neuropathy compared to non-neuropathic subjects. These studies mainly used high-end or not so portable infrared (IR) cameras or contact temperature sensors to measure the skin temperature. However, they assessed skin temperature in steady-state condition, showing only the current state of the skin temperature, not showing the capability of the autonomic system to respond to changing environmental conditions. Some studies applied a cold pressor test (CPT) or an inspiratory gasp to trigger an autonomic response and measured the temperature before and after this intervention. None of these have covered the complete STR curve following the intervention. The shape of this STR curve might be more informative about the state of ANF.

Our project aims to develop a low-cost, portable method that uses IR video thermography to measure accurately the skin temperature of human hands before and after applying CPT at regular intervals during temperature recovery. This study describes the performance in technical terms, and the temperature response variability and reproducibility of healthy subjects.

## Methods

### Measurements and key metrics

The skin temperature for 12 different Regions of Interest (ROI) located on a subject's hands was measured. The measurements consist of roughly 3 phases (see Fig 1):

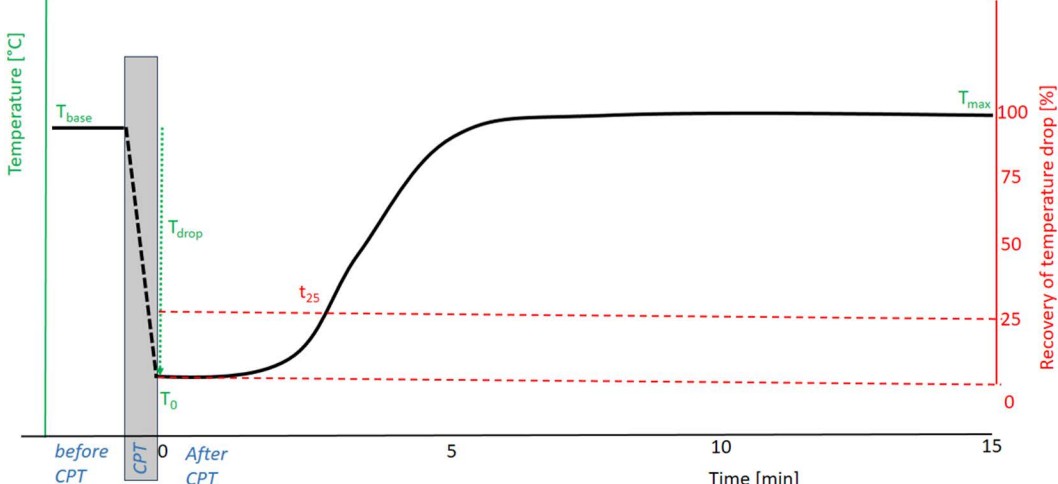

**Fig 1. Metrics defined for STR quantification shown on a hypothetical, typical STR curve.** Green relates to measurements in absolute temperature, red relates to measurements relative to the temperature drop.

1. Baseline skin temperature ($T_{base}$) measurement at each ROI averaged over a 30 second time interval just before applying CPT, after an initial 15 minutes of acclimatization in an air conditioned room maintained at 24°C.

2. Cooling (CPT) during which the subject submerges his/her hands up to the wrists into a container of 5°C cold water for 1 minute to trigger an autonomic response.

3. STR measurement for each ROI measured during 15 minutes of recovery, which is sufficient for recovery of most healthy controls [23].

   • $T_0$: skin temperature immediately after removing CPT.

   • $T_{drop}$: difference between $T_{base}$ and $T_0$.

   • $T_{max}$: maximum temperature reached at end of recovery, averaged over the last 5 samples.

   • $T_{25}$: duration at which 25% of $T_{drop}$ has been recovered.

**Requirements**

The following requirements for the technical setup was defined:

• IR video recording function, to allow taking samples/frames at fixed intervals during post-processing. This provided flexibility to add/change ROI during post-processing, and the option to deploy other image processing techniques if necessary

• 12 ROI placed on both hands, selected during post-processing to have flexibility of changing the data analysis approach in case findings illustrate a different set of ROIs.

• 1 ROI located on the background for reading the background temperature.

• Precise (less than 0.5°C) and accurate (2°C or better) skin temperature recording to allow systemic error correction (calibration) afterwards.

- Affordable, portable hardware, max USD 250.

- Potential to do the full analysis from the chosen hardware platform.

## Camera selection

Three UNI-T UTi721M IR cameras for Android were selected. At the time of purchase these cameras provided the best specifications among the available low-cost infrared cameras. This is a small, affordable IR camera (at the time of selecting and purchasing: EUR 181, September 2023). The camera fits into an Android based mobile phone through a USB-C connection. The Android version was chosen as this is the most common, affordable phone system that many people already have. The camera has a resolution of 256x92 pixels, a thermal sensitivity less than 50mK, and a spatial resolution of 3.8 mrad. It can record both pictures, and videos at a maximum of 25 Hz frame rate. The emissivity can be set, as can the distance between camera and object, and the color map be set. The accuracy is +/-2% or +/- 2°C (whichever is greater) across its range of −20–500°C when used in industrial mode.

## Data extraction method from IR video

**Python script to extract video to numeric data.** A Python script (Python 3.11.5, Anaconda environment, Spyder 5.5.0 interface) was developed to extract the recorded skin temperatures from the IR videos. The script provides a user interface that allows the user to select a video to be analyzed, and provides sliders to indicate the start and end of both baseline (typical duration 20–30 seconds) and recovery phase (typical duration 15–16 minutes) in the video (Fig 2 Left). The script then extracts individual frames for both baseline and recovery phase at a sample frequency set to 1 Hz. The script selects a frame that has maximum difference in contrast between the mean value of the highest 20% and lowest 20% of pixels in the frame, and presents that frame to the user for ROI selection. The user interface guides the user to select the width of the hand palm by selecting 2 locations at each side of the hand palm. The hand palm width is then used to scale the diameter of a circular shaped ROI to the width of the hand palm. Each ROI will be sized to 1/8th of the hand palm width, which corresponds with approximately half a finger width. Except for the ROI on the hand palm

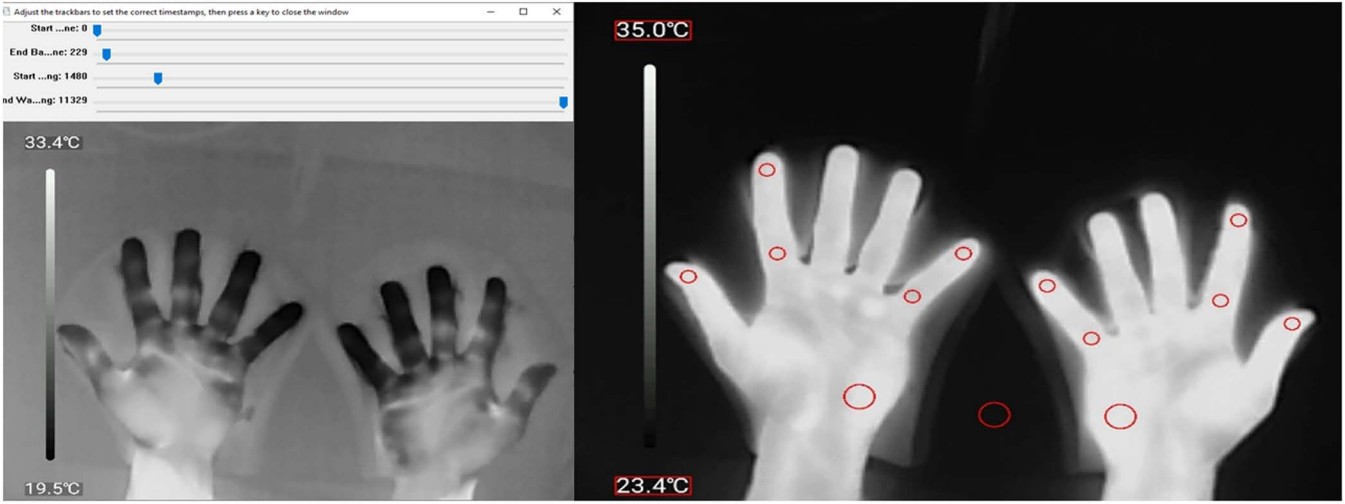

**Fig 2. Temperature images.** Left: Interface to select start and end points of both baseline and recovery phase. Right: Annotated frame with 12 ROI on the hands and 1 ROI on background. Also highlighted are the minimum and maximum value of the temperature scale.

and background, those are sized 1/4$^{th}$ of the hand palm size. In this way, all ROI will have a comparable size between subjects.

Subsequently, the user interface allows selection of 12 ROI within the image, and 1 ROI for reading the background temperature (Fig 2 Right). One frame from the recovery phase is used for ROI selection and then applied to the whole recovery phase. The ROI are also presented on a baseline frame. If the ROI placement is incorrect for the baseline phase, new baseline ROI placement can be given.

The script applies the same baseline and recovery ROI to all extracted frames within baseline and recovery phases, and saves the annotated frames. The script then reads the mean and standard deviation (SD) of all pixel values inside each ROI. The pixel values are converted to temperatures using the minimum and maximum value of the temperature scale that is superimposed on the left side of each frame (Fig 2 Right). Optical Character Recognition (OCR) is used to read the minimum and maximum value in each frame from the indicated boxes (Fig 2 Right). Tesseract is used as a free open-source OCR software. When Tesseract is unable to read the value, a Gaussian blur is applied to improve the outcome. If still unable to read the value, then this frame is skipped from the analysis. For the baseline phase a mean temperature over all extracted baseline frames is calculated for each ROI. For the recovery phase the ROI temperatures are stored for each individual extracted frame. Finally, all data gained from the video is stored in an Excel file that lists all timestamps and ROI temperatures for each frame.

### Matlab script for combining and processing the extracted data

A Matlab script (Matlab R2023b, The Mathworks) was developed to process the data from the individual subject's Excel files. First, each individual subject's Excel file is imported in Matlab. IR camera drift was noticed when plotting the background signal. Introducing cold hands into the picture made the recorded background temperature rise. As background temperature in reality is stable during a short data recording, this background data was used to correct IR camera drift. The error between instantaneous background temperature in each sample and mean background temperature over the first and last 3 measurements was calculated. This error was then subtracted for each sample for all ROI. Next, the temperature data is post-processed by interpolating missing data points to achieve uniform temporal spacing as preparation for filtering. Next, the data is filtered using a low-pass Butterworth filter to reduce noise and artifacts, but still allows to follow a quick recovery response. The filter was designed using the Welch power spectral density estimate of the original signal on temperature response data from 5 healthy subjects. Filter parameters were: 4$^{th}$ order filter, 0.01 Hz normalized cut-off frequency, 1 Hz sample frequency. The filtering was done forwards and backwards using the filtfilt Matlab command to eliminate phase shifts/time delays. The script then plots and outputs the key metrics defined for the research. Focus was on the four most important ROI: distal phalanxes of the left and right index fingers (labeled L2 and R2), and of left and right pinky fingers (labeled L4 and R4). Finally, the combined data of all subjects was stored into one Master Excel file for further statistical analysis

### Setup for measurements

As the Python analysis script relies on steady hand and finger position during the video it was important to secure the hand and finger position for each subject. A setup was developed to ensure steady hands and fingers positions during data (Fig 3). The setup consists of a plywood base plate equipped with Velcro bands, a vertical pole, and 2 molds for hand fixation. Two concave shaped molds were designed to comfortable hold a subject's hands. The molds fixate to the base plate using Velcro, allowing the molds' position and orientation to be adjusted for each subject. The molds are made of PVC, formed into the right shape, and covered with a thin layer of PU foam for increased comfort. Foam finger separators were designed in various sizes to fixate the finger position inside the mold. The finger separators attach to the mold using Velcro allowing finger separator position and orientation to be adapted to each subject. The molds were designed for unobstructed IR camera field of view and unaffected blood supply to the fingers. A 3D-printed camera holder

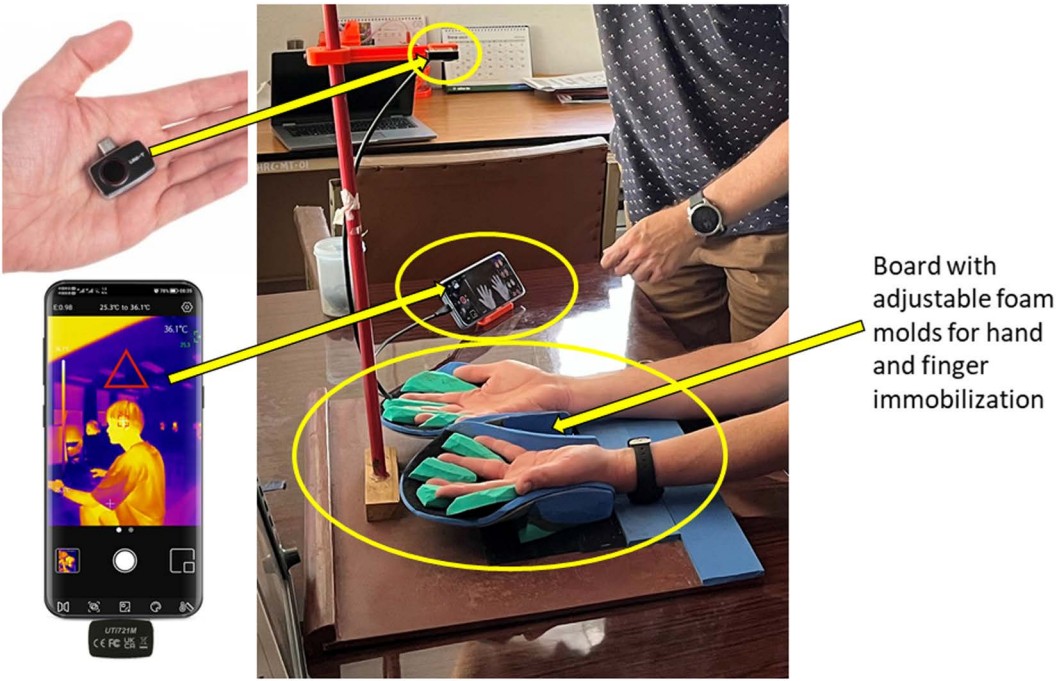

**Fig 3. The hardware setup.** The picture shows the setup consisting of a wooden plate, with 2 hand supports with finger separators, an IR camera in its 3D printed holder mounted on the vertical rod, a mobile phone in its holder, and a subject with his hands positioned in the hand supports.

was designed to connect the IR camera to the vertical stand, and is used to fixate the camera position relative to the base plate. The camera was positioned at 50 cm above the hand palms of a subject's hands fixated into the molded. A 3D printed phone holder was used to ensure correct orientation of the video and temperature scale in each video.

## Verification tests

The technical side of the setup was verified. Several tests were defined to assess the technical performance of the camera in terms of precision, accuracy, and between-camera comparison, to assess the test setup, and to assess the Python script for data extraction.

## Effect of filtering

The effect of filtering was evaluated in 2 ways. Firstly, an IR video recording was made of a container of water in the room during 15 minutes. The video was analyzed by placing a ROI on the water container in the video. The resulting data plotted in one figure showing both unfiltered and filtered data was analyzed. Secondly, the unfiltered and filtered data of one of the subjects was analyzed in a similar way.

## Camera accuracy of skin temperature measurement

To determine the accuracy of the infrared camera measurement an external skin temperature sensor was used as reference. After 15 minutes acclimatization a single subject dipped his left hand for 1 minute in cold water (5°C). The intend of cooling the hand was not to reach a certain reproducible temperature, but to present a skin temperature range to the temperature sensors that is relevant for the clinical application, allowing skin temperature comparison between sensors across this range. After drying the hand the subject had to place his left hand in the measurement setup (Fig 4). A

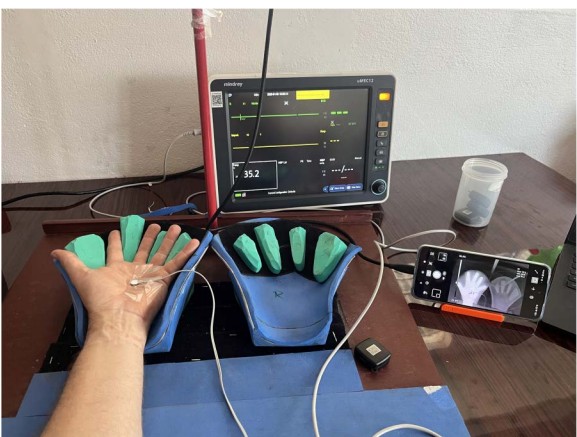

**Fig 4. Test setup for infrared camera accuracy testing. The camera is above the hands, just outside the picture.**

reusable skin temperature sensor connected to a patient monitor (Mindray uMEC 12) was fixated using medical tape at the palm of the left hand, near the center. The sensor temperature was allowed to stabilize. On the thermal camera app (UNI-T Thermal) an approximately 1x1cm rectangle was placed adjacent to the temperature sensor. The app displayed the average temperature within the rectangle (Fig 5). The average IR temperature and the temperature sensor values were recorded every 1 minute while the skin temperature was in the range between 26 and 36°C, or until the hand temperature no longer increased. This recording process was repeated for 3 cameras, with for each single camera repeating 2 times, with disconnecting the IR camera between repetitions to enforce a new calibration. For each camera for 1 run also the video was recorded and temperature data extracted. For this, 1 ROI was placed inside the on-screen rectangle, and 3 other ROI were placed just outside the on-screen rectangle. The 4 resulting temperature series were averaged. The resulting data was processed in Matlab plotting a Bland-Altman plot and intraclass correlation plot, using the patient monitor temperature sensor results as a reference for the IR camera temperatures. These plots were used to assess accuracy and precision of the IR cameras.

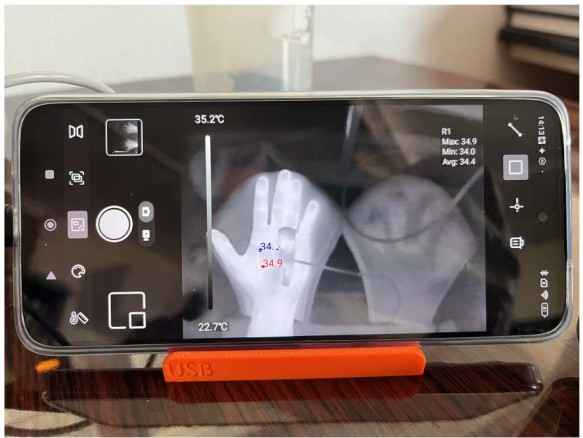

**Fig 5. The phone screen during the accuracy test, showing the rectangle for reading the average temperature that is displayed in the top right corner.**

### Repeatability of data extraction

One subject was enrolled in an initial pilot test on November $2^{nd}$ 2024. Written informed consent was given. The video was analyzed by 2 observers 5 times each to see the reproducibility of the data extraction process using the Python script and to see the effect of the observer analyzing the video. The temperature data was further processed in Matlab. The resulting STR curves for the 4 key ROI (L2, R2, L4, R4) were extracted, and the mean and SD was calculated for each key metric ($T_{base}$, $T_0$, $T_{drop}$, $T_{max}$, and $t_{25}$) and displayed in a figure.

### Repeatability of subject responses

To assess the repeatability of STR measurements, 7 subjects were enrolled to take the CPT after giving written informed consent. Ethical approval was obtained from the Nepal Health Research Council, registration number 81_2024. Enrollment was done from November $11^{th}$ to $15^{th}$ 2024. For each subject, the test was repeated in 5 consecutive days, taking the daily test around the same time. Blood pressure and pulse were recorded before and after CPT. The resulting videos were analyzed and the key metrics were calculated.

## Results

### Effect of filtering

Fig 6 top panel shows a recording of the room temperature water container temperature reading in time, showing the raw, unfiltered and the filtered data. The filter is effectively removing noise but still following the slower trends in the signal.

Fig 6 bottom panel shows a recording of the response of a healthy subject after CPT. Only 1 ROI is shown for clarity. Raw data and filtered data are shown. The filtered signal traces the raw signal without following the noise, and artefacts caused by minor finger movement. It can be noted that high peaks in the raw signal (due to minimum/maximum scale reading mistakes by the OCR) are filtered out, although a high density of high peaks causes the filtered signal to drift away, explaining the sudden bump around 11 min. This is an extreme case of such event.

### Camera accuracy of skin temperature measurement

Fig 7 shows the results of the comparison of the three IR camera readings versus Mindray skin temperature sensor readings as reference temperature. The left panel displays the Intra-Class Correlation plot between the Mindray skin temperature sensor readings and the IR camera readings. For Camera 1, 2 and 3 the mean ICC was 0.98 (SD 0.068), 0.92 (SD < 0.001), and 0.92 (SD 0.035), respectively. Variation (+/-1.96 SD) was between −0.28°C and +0.96°C, −0.060°C and +1.30°C, and −0.10°C and −1.50°C, and the systematic error was + 0.34°C, + 0.64°C, and −0.82°C, respectively. The mean ICC across all cameras was 0.94 (SD 0.035), variation was between −1.30°C and +1.50°C, and the systematic error across was + 0.090°C.

Fig 8 shows how the temperatures extracted from the videos using the Python script correspond with the mean temperature values read from the phone screen. Across 3 cameras the systematic error was −0.48°C. Variation was between −1.40°C and +0.39°C. The mean ICC was 0.88 (SD 0.078).

### Repeatability of data extraction

Fig 9 shows the key metrics of the STR curve during the data extraction repeatability test. For each metric the mean and standard deviation across the 5 repetitions is shown, per observer. The biggest difference can be found in metrics $T_0$, $T_{drop}$, and $t_{25}$.

### Repeatability of subject responses

Fig 10 Top panel shows the measured STR curves in one ROI, for 6 subjects, each measured 5 times. One subject was excluded as the subject could not participate in all 5 repetitions. Clearly, the recovery response looked

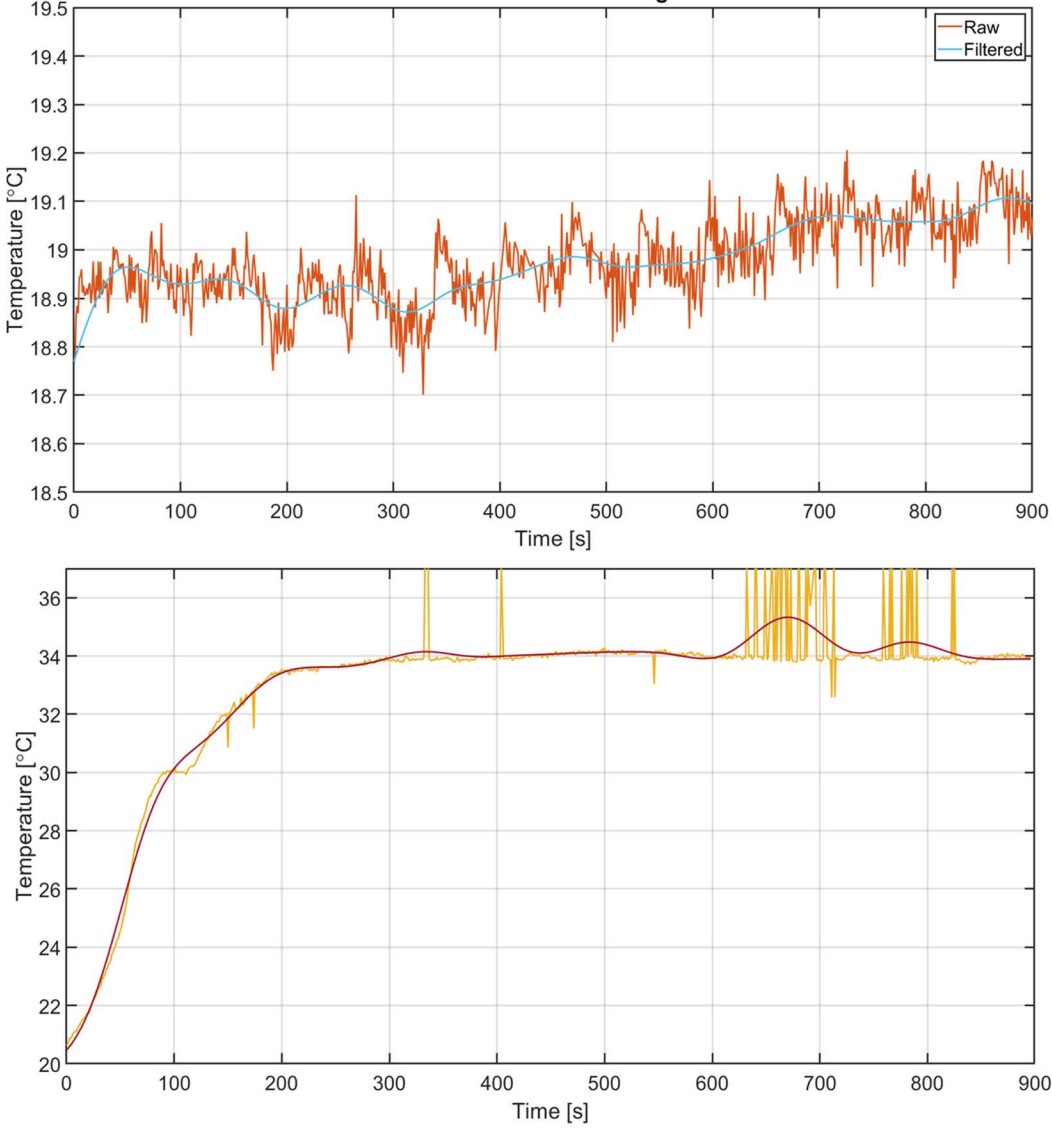

**Fig 6. Effect of filtering.** Top: Filter performance on steady background temperature reading showing the Raw and Filtered signal. Bottom: Filtering on a healthy subject's response, showing the Raw and Filtered signal. The big spikes in the raw signal are caused by misreading of the minimum or maximum temperature scale value by the OCR.

very different for the same individual on a different day. The general shape looks often identical, starting with a slow increase in temperature, followed by a steeper increase in temperature. Especially the moment at which this steep increase occurs varied widely among the subjects and among the repetitions. No relation to blood pressure or pulse could be found.

Fig 10 Bottom panel shows the key metrics $T_{base}$, $T_0$, $T_{max}$, $T_{drop}$ and $t_{25}$ that were defined to describe key moments in the STR curve. Only one ROI, identical to the ROI of Fig 10 Top panel, is included. Other ROIs show similar behavior. Variations in temperature are generally within +/- 1°C. Variations in time are much bigger, often showing +/- 1 min differences for the time to reach 25% recovery.

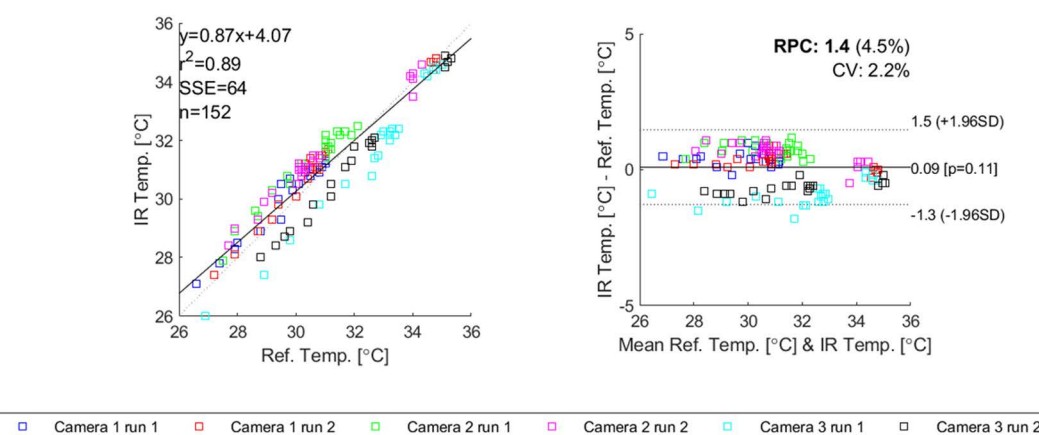

**Fig 7. Combined correlation plot (left) and Bland-Altman plot (right) of all 3 cameras each having 2 repetitions.**

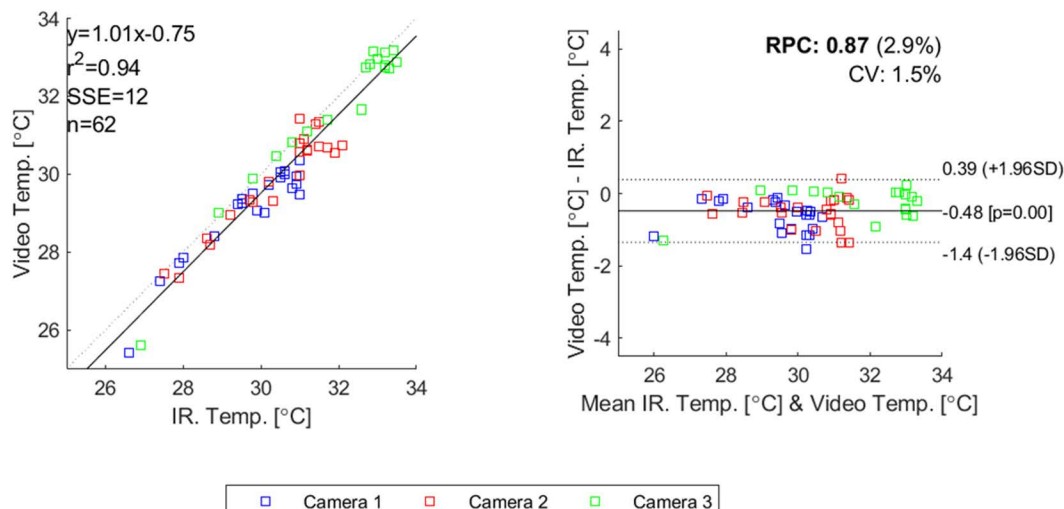

**Fig 8. Combined correlation plot (left) and Bland-Altman plot (right) of all 3 cameras against temperature extracted from video.**

## Discussion

The goal of this study was to develop a low-cost, portable method that uses IR video thermography to measure the human skin temperature before and after a CPT at regular intervals during temperature recovery. A low-cost IR thermal camera that can connect to a smartphone was selected. A setup was made to fixate the camera and hands relative to each other. A protocol was developed to apply a CPT and measure the resulting STR. A Python script was made to extract the temperatures of 12 ROIs in the human hand from recorded infrared videos. A Matlab script was made to post-process the data of individual subjects, and calculate key metrics to describe the subject's response. This method was evaluated on technical performance and in a repeatability test with 6 subjects.

The implemented low-pass filter showed to be effective in reducing noise from the IR image, and in suppressing out-liers caused by misreading of the temperature scale in the IR image. At the same time, the filter still allowed the filtered signal to follow steep sections of the response well. In this case, there are many peaks because of misreading of the

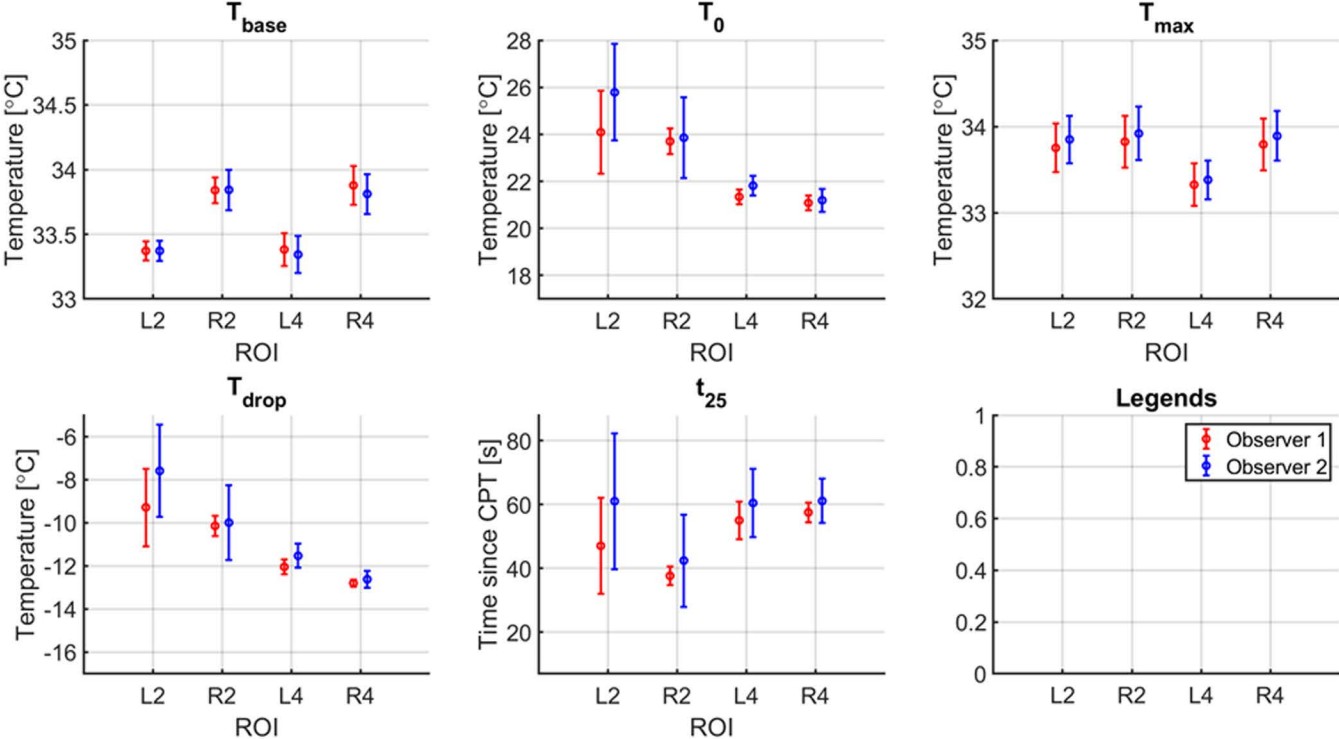

**Fig 9. Repeatability of data extraction on some key metrics.** 5 repetitions on the same video by 2 observers. L2, R2, L4 and R4 correspond with ROI located at the distal phalanx of the left index finger, right index finger, left pinky finger, and right pinky finger, respectively.

minimum or maximum values of the temperature scale, with gaps in between caused by missing data where the minimum or maximum scale could not be read at all. As data is interpolated to ensure 1s sample time, these gaps between misreadings are then filled using interpolation, so effectively the gap is filled with misreadings. This causes the filtered signal to drift away from the raw signal. For future larger-scale data collection, this has to be quantified to estimate the impact of this phenomenon.

The accuracy and precision of the IR camera seems good enough for the application. The high ICC shows good correlation between the IR camera and reference temperature. The IR cameras had a low systematic error, and a good precision according to the Bland-Altman plots. Also the temperature as read from the video compared to the temperature from the patient monitor skin temperature sensor showed low systematic errors, and a good precision. It should be noted that locations of the patient monitor temperature sensor, and the IR on-screen rectangle, and the video extraction ROI were not exactly in the same location. As the hand was recovering from cooling down, the measurement was dynamic, so the measurements needed to be taken at the same instant of time. Physically it was not possible to read all 3 sensors at the same instant of time and in the same location. Therefore, they were positioned as near to each other as was possible. As can be seen from Fig 5, the temperature within the hand palm is not uniform. Therefore, the differences between the various sensors might not have originated from accuracy or precision issues but from actual physical differences in skin temperature between the 3 locations where the measurements were taken. The accuracy and precision results obtained in this study can therefore be seen as a worst case comparison, and the actual accuracy and precision might be much better. Further work, for instance performing a cross-sensor validation test could clarify this issue. Still readings between cameras are expected to be comparable.

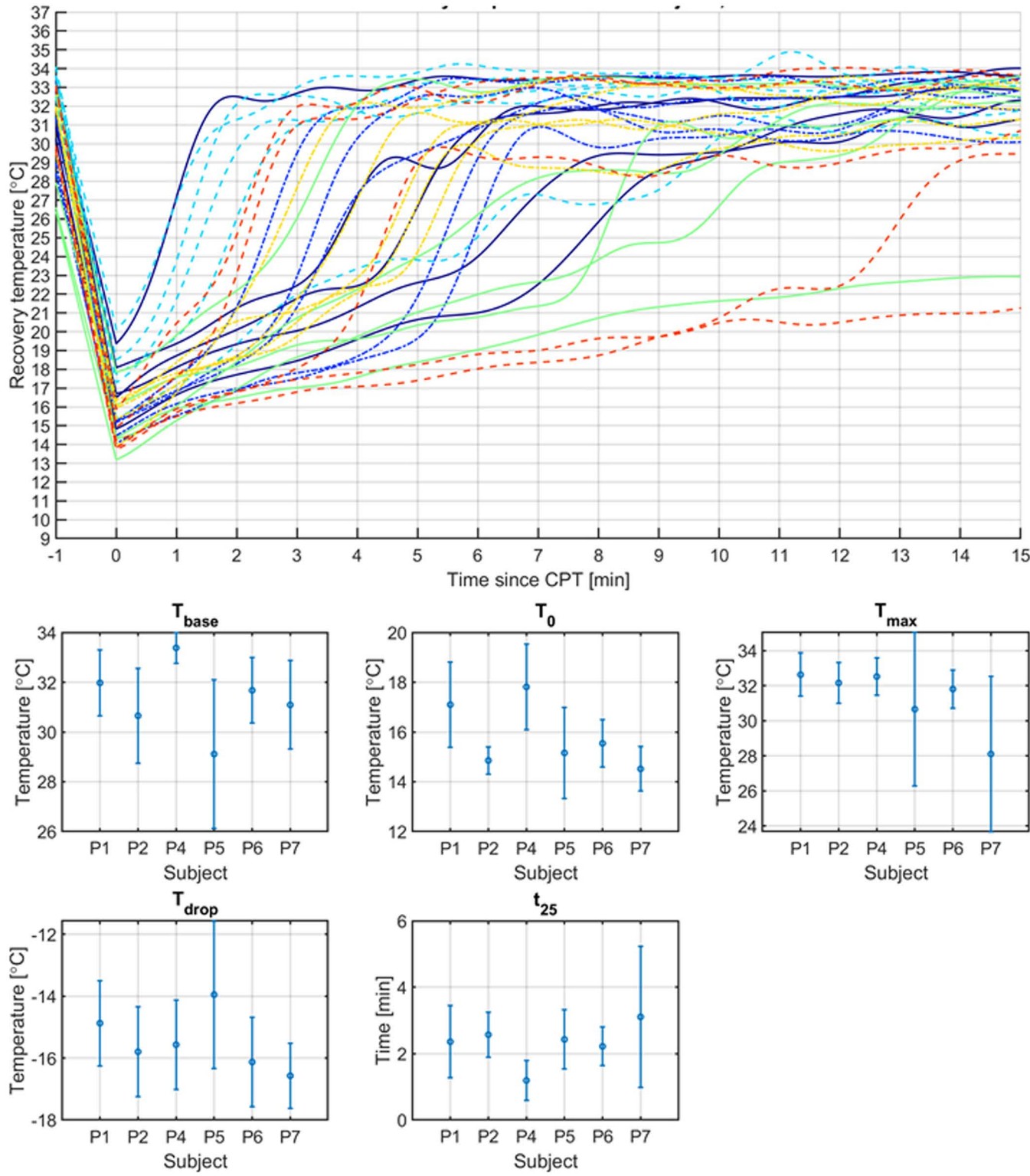

**Fig 10. Temperature variability results of 1 ROI for 5 repeated measurements over 6 subjects.** Top panel: Temperature STR curves grouped by unique color and line style combination per subject. The instance of t = −1 minute indicates the baseline temperature before applying CPT. The instance t = 0 indicates $T_o$, the temperature direct after applying CPT. Bottom panel: The resulting key metrics (mean and SD) extracted from the STR curves. Subject P3 was excluded as this subject did only do 3 repetitions instead of 5.

An alternative could have been to use a temperature controlled object to take the measurements from. However, since IR cameras rely on the emissivity coefficient of the object they are observing to estimate the object's temperature the emissivity coefficient of the object needs to be similar to the emissivity of human skin. Since we did not have any temperature controlled object made from a material with identical emissivity coefficient as the human skin it was not possible to take this approach. A future study might explore the accuracy and precision of the adopted skin temperature measurement further.

The inter- and intra-observer repeatability of the video data extraction process was high. Between observers the largest differences were found for $T_0$ and $t_{25}$. A closer inspection of the subject in question showed that the subject placed his hands back in the holders and maintained a steady position for a few seconds, then adjusted his hands, and then kept a steady hand position until the end of the recording. It seemed that 1 observer chose the first steady hand position as start of the recovery phase, while the other observer chose the second steady hand position. This explains both the quite large difference in $t_{25}$, as the relatively large difference between $T_0$ and $T_{drop}$, which are all strongly related to the chosen start of the recovery phase. $T_{base}$ and $T_{max}$ are not depending on this moment, and therefore these moments showed a very low variation. A potential approach to solve this issue could be the deployment of a machine learning model that identifies the ROI automatically.

The variability of a subject's response over repeated measurements was bigger than expected. The subjects had 15 minutes to acclimatize in a temperature controlled room, and measured at the same time of the day. The expectation was a highly similar STR curve for any subject every time. The subjects generally followed the expected STR curve shape, but the timings of the various regimes of the STR curve were different. It suggests that other factors besides the controlled factors room temperature, time of the day, and acclimatization are influencing the STR. No relation could be found with blood pressure or pulse. It might be that previous activities on that day, like exercise, or coffee consumption were causing the autonomic system or the blood supply to the hands to be in a different state. The variation in STR curve shapes also implies that a single measurement of a subject might not be sufficient to have solid conclusions regarding when determining, e.g., ANF, unless affected patients show a fundamentally different behavior that can be easily discerned from the variations among healthy subjects. Previous studies looking into skin temperature (either at baseline, or after cold stimuli, or even measuring the full recovery) as an indicator for ANF have not taken this within subject variation into account. A future study including both healthy subjects and leprosy-affected patients will be needed to provide more information about the differences in skin temperatures.

## Conclusion

This study shows that a low-cost, portable IR camera can be used to measure skin temperature response of human hands. A measurement setup was developed to measure skin temperature before and after applying cold pressor test and determine the following STR curve at 1s intervals using the IR camera's video function. Custom software was made to extract the temperature in 12 Regions of Interest from recorded videos, and metrics were defined to describe the resulting STR curve. The method presented showed high intraclass correlation (ICC > 0.9) between 3 IR cameras and a medical skin temperature sensor that was used as reference. The mean accuracy over the 3 cameras was +0.090°C, and the variation (+/- 1.96SD) was between −1.30 and +1.50°C. A pilot test on healthy subjects showed that healthy subjects show high variability in their response when testing the subjects repeatedly. This implicates single measurements per subject might give unreliable readings of the STR curve, unless affected subjects have a fundamentally different STR curve. Future research with affected subjects having, e.g., ANF and healthy controls will give more insight into this.

## Author contributions

**Conceptualization:** Arjan J. Knulst, Fleur van den Bogaert, Alexander Kuipers, Lieke Roelofs, Corine Knulst-Verlaan.

**Data curation:** Arjan J. Knulst, Fleur van den Bogaert, Alexander Kuipers, Lieke Roelofs.

**Formal analysis:** Arjan J. Knulst.

**Funding acquisition:** Arjan J. Knulst, Jenny Dankelman, Wim Brandsma, Corine Knulst-Verlaan, Suraj Maharjan.

**Investigation:** Arjan J. Knulst, Fleur van den Bogaert, Alexander Kuipers, Lieke Roelofs.

**Methodology:** Arjan J. Knulst, Fleur van den Bogaert, Alexander Kuipers, Lieke Roelofs.

**Project administration:** Arjan J. Knulst, Suraj Maharjan.

**Resources:** Arjan J. Knulst, Jenny Dankelman, Suraj Maharjan.

**Software:** Fleur van den Bogaert, Alexander Kuipers, Lieke Roelofs.

**Supervision:** Arjan J. Knulst.

**Validation:** Corine Knulst-Verlaan.

**Visualization:** Arjan J. Knulst, Fleur van den Bogaert, Alexander Kuipers, Lieke Roelofs.

**Writing – original draft:** Arjan J. Knulst.

**Writing – review & editing:** Jenny Dankelman, Wim Brandsma, Corine Knulst-Verlaan, Suraj Maharjan.

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
