## [Decision Letter · Decision Letter 0]

PONE-D-25-12017Pre-clinical evaluation of a low-cost tool for skin temperature measurements to assess autonomic nerve function impairment.PLOS ONE

Dear Dr. Knulst,

Thank you for submitting your manuscript to PLOS ONE. After careful consideration, we feel that it has merit but does not fully meet PLOS ONE’s publication criteria as it currently stands. Therefore, we invite you to submit a revised version of the manuscript that addresses the points raised during the review process.

We look forward to receiving your revised manuscript.

Kind regards,

Iftikhar Ahmed Khan

Academic Editor

PLOS ONE

Journal Requirements:

Reviewers' comments:

Reviewer's Responses to Questions

**Comments to the Author**

1. Is the manuscript technically sound, and do the data support the conclusions?

Reviewer #1: Yes

Reviewer #2: Yes

2. Has the statistical analysis been performed appropriately and rigorously? 

Reviewer #1: Yes

Reviewer #2: Yes

3. Have the authors made all data underlying the findings in their manuscript fully available?

Reviewer #1: Yes

Reviewer #2: Yes

4. Is the manuscript presented in an intelligible fashion and written in standard English?

Reviewer #1: Yes

Reviewer #2: Yes

5. Review Comments to the Author

Reviewer #1: General Comments

1. The findings of this study show that a low-cost IR camera can be used to measure STR. However, these results alone do not substantiate the claim in the title that this low-cost tool can assess autonomic nerve function impairment. Although the future research in this direction is mentioned in the conclusion, the current study does not establish a direct relationship between skin temperature and autonomic nerve function impairment. Therefore, additional experiments specifically designed to explore this relationship should be incorporated into the main body of the paper.

2. In the discussion section, it is mentioned that the observed differences among various sensors might not only stem from issues related to accuracy or precision, but also from actual physical differences in skin temperature across the three measurement locations. Employing a cross-sensor validation test could effectively address this issue.

Specific Comments

1. In the manuscript, it is mentioned that the resulting STR curve should be recorded for 15 minutes. Why choose 15 minutes? Why not 10 minutes or 20 minutes?

2. Why use an IR video camera? What are the advantages of an IR video camera over contact temperature sensors? There are many low-cost portable contact temperature sensors could be chosen.

3. Why use a Python script to extract video to numeric data and use a MATLAB script for combining and processing the extracted data? Why use the same language?

4. In the third paragraph of the abstract, the last sentence “both within as between…” is inappropriate.

5. In the methods part, how to quantify the submersion depth of subjects’ hands to guarantee repeatability?

6. In the Camera selection part, the expression of ‘EUR 2000 – 18.000’ is incorrect.

7. The expression of ‘warming phase’ of Figure 2 is not mentioned in other parts, does it mean ‘recovery phase’?

8. The IR camera position in the left picture of Figure 3 is not clear, so I highly recommend using distinctive curves and arrows to indicate the position.

9. In the part of Camera accuracy of skin temperature measurement, how to guarantee the same temperature changes when drying the hand before placing the hand in the measurement setup among different repeated experiments?

10. In the part of Camera accuracy of skin temperature measurement, it is mentioned that … until the hand temperature had stabilized. What is the criterion for hand temperature stabilization?

11. In the part of Camera accuracy of skin temperature measurement, the expression of ‘The left panel displays the Intra-Class Correlation plot between the two’ is not clear.

12. In the first paragraph of the discussion, the expression of ‘A Python script was made to extract the temperatures of 10 ROIs…’ is not in accordance with the 12 ROIs in other parts.

13. In the manuscript, it is mentioned that between observers, the largest differences were found for T0 and t25. Can script-based automated recognition replace manual work?

Reviewer #2: This is a well-designed study. It's believed the proposed low-cost experimental setup can not only be used for assessment of peripheral autonomic nerve function but also for assessment of microcirculation function. The reviewer has only several small suggestions

1. The low-cost IR camera is the key instrument. One picture showing the camera and the connection to the mobile phone may be presented in Fig. 4 beside the fixing supporter and the mobile phone.

2. The designed fixing suporter is a very good apparatus for avoiding hand moving. But in some circumstances the temperatures of the supporter may be close to the finger temperature �for example, in Fig. 5 the boundary between the 3rd and 4th finger seems to be a little bit blurred�. The method to identify the finger boundary may be explained more in details.

3. In the introduction part, it's mentioned that " various studies [17–22] have been done, mainly using high-end or not so portable infrared (IR) cameras, or contact temperature sensors". Here� more detailed descriptions may be given to show how the skin temperature variations are associated with ANF. Similarly, in discussion part, it's written that "other factors besides room temperature, time of the day, and acclimatization are influencing the outcomes". It's considered that skin temperature is closely related to skin blood flow and metabolic heat generation. Here, the discusion may be more extended.

6. PLOS authors have the option to publish the peer review history of their article (what does this mean? ). If published, this will include your full peer review and any attached files.

**Do you want your identity to be public for this peer review?** For information about this choice, including consent withdrawal, please see our Privacy Policy .

Reviewer #1: **Yes: ** Chenxi Yang

Reviewer #2: No

---

## [Author Response · Author response to Decision Letter 1]

6 Jun 2025

We are thankful to the reviewers to help us improve the readability and quality of the paper. We expect to have addressed all comments in our revised version. See our response to the reviewers document, and see the revised manuscript.

Kind regards

Arjan J. Knulst

---

## [Decision Letter · Decision Letter 1]

Pre-clinical evaluation of a low-cost tool for skin temperature measurements as a proxy to assess autonomic nerve function in leprosy neuropathy.

PONE-D-25-12017R1

Dear Dr. Knulst,

We’re pleased to inform you that your manuscript has been judged scientifically suitable for publication and will be formally accepted for publication once it meets all outstanding technical requirements.

Kind regards,

Iftikhar Ahmed Khan

Academic Editor

PLOS ONE

Additional Editor Comments (optional):

Reviewers' comments:

Reviewer's Responses to Questions

**Comments to the Author**

1. If the authors have adequately addressed your comments raised in a previous round of review and you feel that this manuscript is now acceptable for publication, you may indicate that here to bypass the “Comments to the Author” section, enter your conflict of interest statement in the “Confidential to Editor” section, and submit your "Accept" recommendation.

Reviewer #1: All comments have been addressed

2. Is the manuscript technically sound, and do the data support the conclusions?

Reviewer #1: Yes

3. Has the statistical analysis been performed appropriately and rigorously? 

Reviewer #1: Yes

4. Have the authors made all data underlying the findings in their manuscript fully available?

Reviewer #1: Yes

5. Is the manuscript presented in an intelligible fashion and written in standard English?

Reviewer #1: Yes

6. Review Comments to the Author

Reviewer #1: The authors have addressed my concerns and the questions from other reviewers. I don't have any further questions.

7. PLOS authors have the option to publish the peer review history of their article (what does this mean? ). If published, this will include your full peer review and any attached files.

**Do you want your identity to be public for this peer review?** For information about this choice, including consent withdrawal, please see our Privacy Policy .

Reviewer #1: No

---

## [Editor Report · Acceptance letter]

PONE-D-25-12017R1

PLOS ONE

Dear Dr. Knulst,

I'm pleased to inform you that your manuscript has been deemed suitable for publication in PLOS ONE. Congratulations! Your manuscript is now being handed over to our production team.

Kind regards,

on behalf of

Dr. Iftikhar Ahmed Khan

Academic Editor

PLOS ONE